# NVAE: A Deep Hierarchical Variational Autoencoder

**Arash Vahdat, Jan Kautz**
NVIDIA
{avahdat, jkautz}@nvidia.com

## Abstract

Normalizing flows, autoregressive models, variational autoencoders (VAEs), and deep energy-based models are among competing likelihood-based frameworks for deep generative learning. Among them, VAEs have the advantage of fast and tractable sampling and easy-to-access encoding networks. However, they are currently outperformed by other models such as normalizing flows and autoregressive models. While the majority of the research in VAEs is focused on the statistical challenges, we explore the orthogonal direction of carefully designing neural architectures for hierarchical VAEs. We propose Nouveau VAE (NVAE), a deep hierarchical VAE built for image generation using depth-wise separable convolutions and batch normalization. NVAE is equipped with a residual parameterization of Normal distributions and its training is stabilized by spectral regularization. We show that NVAE achieves state-of-the-art results among non-autoregressive likelihood-based models on the MNIST, CIFAR-10, CelebA 64, and CelebA HQ datasets and it provides a strong baseline on FFHQ. For example, on CIFAR-10, NVAE pushes the state-of-the-art from 2.98 to 2.91 bits per dimension, and it produces high-quality images on CelebA HQ as shown in Fig. 1. To the best of our knowledge, NVAE is the first successful VAE applied to natural images as large as 256×256 pixels. The source code is available at https://github.com/NVlabs/NVAE.

## 1 Introduction

The majority of the research efforts on improving VAEs [1, 2] is dedicated to the statistical challenges, such as reducing the gap between approximate and true posterior distributions [3, 4, 5, 6, 7, 8, 9, 10], formulating tighter bounds [11, 12, 13, 14], reducing the gradient noise [15, 16], extending VAEs to discrete variables [17, 18, 19, 20, 21, 22, 23], or tackling posterior collapse [24, 25, 26, 27]. The role of neural network architectures for VAEs is somewhat overlooked, as most previous work borrows the architectures from classification tasks.

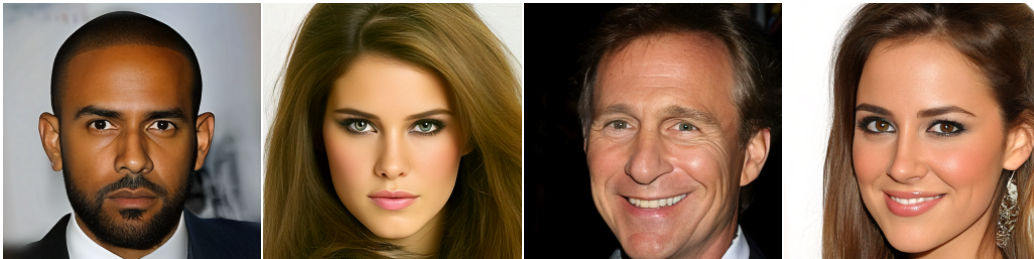

Figure 1: 256×256-pixel samples generated by NVAE, trained on CelebA HQ [28].

However, VAEs can benefit from designing special network architectures as they have fundamentally different requirements. First, VAEs maximize the mutual information between the input and latent variables [29, 30], requiring the networks to retain the information content of the input data as much

as possible. This is in contrast with classification networks that discard information regarding the input [31]. Second, VAEs often respond differently to the over-parameterization in neural networks. Since the marginal log-likelihood only depends on the generative model, overparameterizing the decoder network may hurt the test log-likelihood, whereas powerful encoders can yield better models because of reducing the amortization gap [6]. Wu et al. [32] observe that the marginal log-likelihood, estimated by non-encoder-based methods, is not sensitive to the encoder overfitting (see also Fig. 9 in [19]). Moreover, the neural networks for VAEs should model long-range correlations in data [33, 34, 35], requiring the networks to have large receptive fields. Finally, due to the unbounded Kullback–Leibler (KL) divergence in the variational lower bound, training very deep hierarchical VAEs is often unstable. The current state-of-the-art VAEs [4, 36] omit batch normalization (BN) [37] to combat the sources of randomness that could potentially amplify their instability.

In this paper, we aim to *make VAEs great again* by architecture design. We propose Nouveau VAE (NVAE), a deep hierarchical VAE with a carefully designed network architecture that produces high-quality images. NVAE obtains the state-of-the-art results among non-autoregressive likelihood-based generative models, reducing the gap with autoregressive models. The main building block of our network is depthwise convolutions [38, 39] that rapidly increase the receptive field of the network without dramatically increasing the number of parameters.

In contrast to the previous work, we find that BN is an important component of the success of deep VAEs. We also observe that instability of training remains a major roadblock when the number of hierarchical groups is increased, independent of the presence of BN. To combat this, we propose a residual parameterization of the approximate posterior parameters to improve minimizing the KL term, and we show that spectral regularization is key to stabilizing VAE training.

In summary, we make the following contributions: i) We propose a novel deep hierarchical VAE, called NVAE, with depthwise convolutions in its generative model. ii) We propose a new residual parameterization of the approximate posteriors. iii) We stabilize training deep VAEs with spectral regularization. iv) We provide practical solutions to reduce the memory burden of VAEs. v) We show that deep hierarchical VAEs can obtain state-of-the-art results on several image datasets, and can produce high-quality samples even when trained with the original VAE objective. To the best of our knowledge, NVAE is the first successful application of VAEs to images as large as $256 \times 256$ pixels.

**Related Work:** Recently, VQ-VAE-2 [40] demonstrated high-quality generative performance for large images. Although VQ-VAE's formulation is motivated by VAEs, its objective does not correspond to a lower bound on data log-likelihood. In contrast, NVAE is trained directly with the VAE objective. Moreover, VQ-VAE-2 uses PixelCNN [41] in its prior for latent variables up to $128 \times 128$ dims that is very slow to sample from, while NVAE uses an unconditional decoder in the data space.

Our work is related to VAEs with inverse autoregressive flows (IAF-VAEs) [4]. NVAE borrows the statistical models (i.e., hierarchical prior and approximate posterior, etc.etc) from IAF-VAEs. But, it differs from IAF-VAEs in terms of i) neural networks implementing these models, ii) the parameterization of approximate posteriors, and iii) scaling up the training to large images. Nevertheless, we provide ablation experiments on these aspects, and we show that NVAE outperform the original IAF-VAEs by a large gap. Recently, BIVA [36] showed state-of-the-art VAE results by extending bidirectional inference to latent variables. However, BIVA uses neural networks similar to IAF-VAE, and it is trained on images as large as $64 \times 64$ px. To keep matters simple, we use the hierarchical structure from IAF-VAEs, and we focus on carefully designing the neural networks. We expect improvements in performance if more complex hierarchical models from BIVA are used. Early works DRAW [5] and Conv DRAW [42] use recurrent neural networks to model hierarchical dependencies.

## 2 Background

Here, we review VAEs, their hierarchical extension, and bidirectional encoder networks [4, 43].

The goal of VAEs [1] is to train a generative model in the form of $p(\boldsymbol{x}, \boldsymbol{z}) = p(\boldsymbol{z})p(\boldsymbol{x}|\boldsymbol{z})$ where $p(\boldsymbol{z})$ is a prior distribution over latent variables $\boldsymbol{z}$ and $p(\boldsymbol{x}|\boldsymbol{z})$ is the likelihood function or decoder that generates data $\boldsymbol{x}$ given latent variables $\boldsymbol{z}$. Since the true posterior $p(\boldsymbol{z}|\boldsymbol{x})$ is in general intractable, the generative model is trained with the aid of an approximate posterior distribution or encoder $q(\boldsymbol{z}|\boldsymbol{x})$.

In deep hierarchical VAEs [5, 9, 4, 43, 44], to increase the expressiveness of both the approximate posterior and prior, the latent variables are partitioned into disjoint groups, $\boldsymbol{z} = \{\boldsymbol{z}_1, \boldsymbol{z}_1, \ldots, \boldsymbol{z}_L\}$,

where $L$ is the number of groups. Then, the prior is represented by $p(\boldsymbol{z}) = \prod_l p(\boldsymbol{z}_l|\boldsymbol{z}_{<l})$ and the approximate posterior by $q(\boldsymbol{z}|\boldsymbol{x}) = \prod_l q(\boldsymbol{z}_l|\boldsymbol{z}_{<l},\boldsymbol{x})$ where each conditional in the prior ($p(\boldsymbol{z}_l|\boldsymbol{z}_{<l})$) and the approximate posterior ($q(\boldsymbol{z}_l|\boldsymbol{z}_{<l},\boldsymbol{x})$) are represented by factorial Normal distributions. We can write the variational lower bound $\mathcal{L}_{\text{VAE}}(\boldsymbol{x})$ on $\log p(\boldsymbol{x})$ as:

$$\mathcal{L}_{\text{VAE}}(\boldsymbol{x}) := \mathbb{E}_{q(\boldsymbol{z}|\boldsymbol{x})}\left[\log p(\boldsymbol{x}|\boldsymbol{z})\right] - \text{KL}(q(\boldsymbol{z}_1|\boldsymbol{x})||p(\boldsymbol{z}_1)) - \sum_{l=2}^{L} \mathbb{E}_{q(\boldsymbol{z}_{<l}|\boldsymbol{x})}\left[\text{KL}(q(\boldsymbol{z}_l|\boldsymbol{x},\boldsymbol{z}_{<l})||p(\boldsymbol{z}_l|\boldsymbol{z}_{<l}))\right], \quad (1)$$

where $q(\boldsymbol{z}_{<l}|\boldsymbol{x}) := \prod_{i=1}^{l-1} q(\boldsymbol{z}_i|\boldsymbol{x},\boldsymbol{z}_{<i})$ is the approximate posterior up to the $(l-1)^{th}$ group. The objective is trained using the reparameterization trick [1, 2].

The main question here is how to implement the conditionals in $p(\boldsymbol{x},\boldsymbol{z})$ and $q(\boldsymbol{z}|\boldsymbol{x})$ using neural networks. For modeling the generative model, a top-down network generates the parameters of each conditional. After sampling from each group, the samples are combined with deterministic feature maps and passed to the next group (Fig. 2b). For inferring the latent variables in $q(\boldsymbol{z}|\boldsymbol{x})$, we require a bottom-up deterministic network to extract representation from input $\boldsymbol{x}$. Since the order of latent variable groups are shared between $q(\boldsymbol{z}|\boldsymbol{x})$ and $p(\boldsymbol{z})$, we also require an additional top-down network to infer latent variables group-by-group. To avoid the computation cost of an additional top-down model, in bidirectional inference [4], the representation extracted in the top-down model in the generative model is reused for inferring latent variables (Fig. 2a). IAF-VAEs [4] relies on regular residual networks [45] for both top-down and bottom-up models without any batch normalization, and it has been examined on small images only.

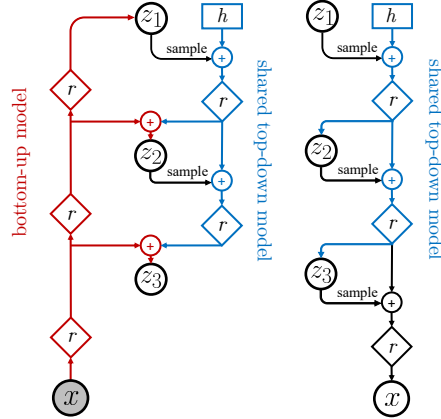

(a) Bidirectional Encoder  (b) Generative Model

Figure 2: The neural networks implementing an encoder $q(\boldsymbol{z}|\boldsymbol{x})$ and generative model $p(\boldsymbol{x},\boldsymbol{z})$ for a 3-group hierarchical VAE. $\diamondsuit$ denotes residual neural networks, $\oplus$ denotes feature combination (e.g., concatenation), and $\boxed{h}$ is a trainable parameter.

## 3 Method

In this paper, we propose a deep hierarchical VAE called NVAE that generates large high-quality images. NVAE's design focuses on tackling two main challenges: (i) designing expressive neural networks specifically for VAEs, and (ii) scaling up the training to a large number of hierarchical groups and image sizes while maintaining training stability. NVAE uses the conditional dependencies from Fig. 2, however, to address the above-mentioned challenges, it is equipped with novel network architecture modules and parameterization of approximate posteriors. Sec. 3.1 introduces NVAE's residual cells. Sec. 3.2 presents our parameterization of posteriors and our solution for stable training.

### 3.1 Residual Cells for Variational Autoencoders

One of the key challenges in deep generative learning is to model the long-range correlations in data. For example, these correlations in the images of faces are manifested by a uniform skin tone and the general left-right symmetry. In the case of VAEs with unconditional decoder, such long-range correlations are encoded in the latent space and are projected back to the pixel space by the decoder.

A common solution to the long-range correlations is to build a VAE using a hierarchical multi-scale model. Our generative model starts from a small spatially arranged latent variables as $\boldsymbol{z}_1$ and samples from the hierarchy group-by-group while gradually doubling the spatial dimensions. This multi-scale approach enables NVAE to capture global long-range correlations at the top of the hierarchy and local fine-grained dependencies at the lower groups.

#### 3.1.1 Residual Cells for the Generative Model

In addition to hierarchical modeling, we can improve modeling the long-range correlations by increasing the receptive field of the networks. Since the encoder and decoder in NVAE are implemented by deep residual networks [45], this can be done by increasing the kernel sizes in the convolutional path.

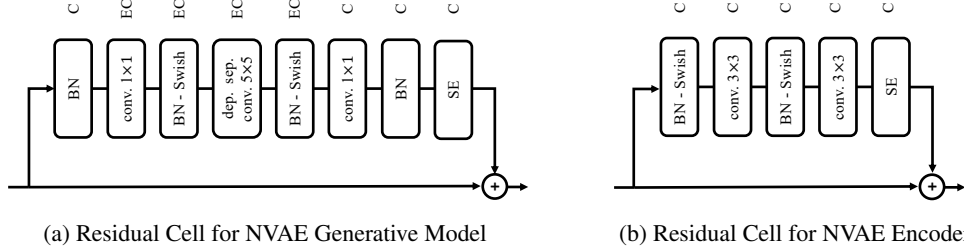

(a) Residual Cell for NVAE Generative Model  (b) Residual Cell for NVAE Encoder

Figure 3: The NVAE residual cells for generative and encoder models are shown in (a) and (b). The number of output channels is shown above. The residual cell in (a) expands the number of channels $E$ times before applying the depthwise separable convolution, and then maps it back to $C$ channels. The cell in (b) applies two series of BN-Swish-Conv without changing the number of channels.

However, large filter sizes come with the cost of large parameter sizes and computational complexity. In our early experiments, we empirically observed that depthwise convolutions outperform regular convolutions while keeping the number of parameters and the computational complexity orders of magnitudes smaller[1]. However, depthwise convolutions have limited expressivity as they operate in each channel separately. To tackle this issue, following MobileNetV2 [46], we apply these convolutions after expanding the number of channels by a $1 \times 1$ regular convolution and we map their output back to original channel size using another $1 \times 1$ regular convolution.

**Batch Normalization:** The state-of-the-art VAE [4, 36] models have omitted BN as they observed that "the noise introduced by batch normalization hurts performance" [4] and have relied on weight normalization (WN) [47] instead. In our early experiments, we observed that the negative impact of BN is during evaluation, not training. Because of using running statistics in BN, the output of each BN layer can be slightly shifted during evaluation, causing a dramatic change in the network output. To fix this, we adjust the momentum hyperparameter of BN, and we apply a regularization on the norm of scaling parameters in BN layers to ensure that a small mismatch in statistics is not amplified by BN. This follows our network smoothness regularization that is discussed in Sec. 3.2.

**Swish Activation:** The Swish activation [48], $f(u) = \frac{u}{1+e^{-u}}$, has been recently shown promising results in many applications [49, 50]. We also observe that the combination of BN and Swish outperforms WN and ELU activation [51] used by the previous works [4, 36].

**Squeeze and Excitation (SE):** SE [52] is a simple channel-wise gating layer that has been used widely in classification problems [49]. We show that SE can also improve VAEs.

**Final cell:** Our residual cells with depthwise convolutions are visualized in Fig. 3(a). Our cell is similar to MobileNetV2 [46], with three crucial differences; It has two additional BN layers at the beginning and the end of the cell and it uses Swish activation function and SE.

### 3.1.2 Residual Cells for the Encoder Model

We empirically observe that depthwise convolutions are effective in the generative model and do not improve the performance of NVAE when they are applied to the bottom-up model in encoder. Since regular convolutions require less memory, we build the bottom-up model in encoder by residual cells visualized in Fig. 3(b). We empirically observe that BN-Activation-Conv performs better than the original Conv-BN-Activation [45] in regular residual cells. A similar observation was made in [53].

### 3.1.3 Reducing the Memory Requirements

The main challenge in using depthwise convolutions is the high memory requirement imposed by the expanded features. To tackle this issue, we use two tricks: (i) We define our model in mixed-precision using the NVIDIA APEX library [54]. This library has a list of operations (including convolutions) that can safely be cast to half-precision floats. This enables us to reduce the GPU memory by 40%. (ii) A careful examination of the residual cells in Fig. 3 reveals that one copy of feature maps for each

operation is stored for the backward pass[2]. To reduce the memory, we fuse BN and Swish and we store only one feature map for the backward pass, instead of two. This trick is known as gradient check-pointing [55, 56] and it requires recomputing BN in the backward pass. The additional BN computation does not change the training time significantly, but it results in another 18% reduction in memory usage for our model on CIFAR-10. These two tricks together help us roughly double the training throughput using a larger batch size (from 34 images/sec to 64 images/sec).

## 3.2 Taming the Unbounded KL Term

In practice, training deep hierarchical VAE poses a great optimization challenge due to unbounded KL from $q(z_l|x, z_{<l})$ to $p(z_l|z_{<l})$ in the objective. It is common to use two separate neural networks to generate the parameters of these distributions. However, in the case of a large number of latent variable groups, keeping these distributions in harmony is very challenging. If the encoder and decoder produce distributions far from each other during training, the sharp gradient update, resulting from KL, will push the model parameters to an unstable region, from which it is difficult to recover. Here, we propose two approaches for improving KL optimization and stabilizing the training.

**Residual Normal Distributions:** We propose a residual distribution that parameterizes $q(z|x)$ relative to $p(z)$. Let $p(z_l^i|z_{<l}) := \mathcal{N}(\mu_i(z_{<l}), \sigma_i(z_{<l}))$ be a Normal distribution for the $i^{th}$ variable in $z_l$ in prior. We define $q(z_l^i|z_{<l}, x) := \mathcal{N}(\mu_i(z_{<l}) + \Delta\mu_i(z_{<l}, x), \sigma_i(z_{<l}) \cdot \Delta\sigma_i(z_{<l}, x))$, where $\Delta\mu_i(z_{<l}, x)$ and $\Delta\sigma_i(z_{<l}, x)$ are the relative location and scale of the approximate posterior with respect to the prior. With this parameterization, when the prior moves, the approximate posterior moves accordingly, if not changed. The benefit of this formulation can be also seen when we examine the KL term in $\mathcal{L}_{VAE}$:

$$\mathrm{KL}\big(q(z^i|x)||p(z^i)\big) = \frac{1}{2}\left(\frac{\Delta\mu_i^2}{\sigma_i^2} + \Delta\sigma_i^2 - \log\Delta\sigma_i^2 - 1\right), \tag{2}$$

where we have dropped subscript $l$ and the dependencies for the ease of notation. As we can see above, if $\sigma_i$, generated by the decoder, is bounded from below, the KL term mainly depends on the relative parameters, generated by the single encoder network. We hypothesize that minimizing KL in this parameterization is easier than when $q(z_l^i|z_{<l}, x)$ predicts the absolute location and scale. With a similar motivation, a weighted averaging of approximate posterior and prior parameters is also introduced in [43].

**Spectral Regularization (SR):** The residual Normal distributions do not suffice for stabilizing VAE training as KL in Eq. 2 is still unbounded. To bound KL, we need to ensure that the encoder output does not change dramatically as its input changes. This notion of smoothness is characterized by the Lipschitz constant. We hypothesize that by regularizing the Lipschitz constant, we can ensure that the latent codes predicted by the encoder remain bounded, resulting in a stable KL minimization.

Since estimating the Lipschitz constant of a network is intractable, we use the SR [57] that minimizes the Lipschitz constant for each layer. Formally, we add the term $\mathcal{L}_{SR} = \lambda \sum_i s^{(i)}$ to $\mathcal{L}_{VAE}$, where $s^{(i)}$ is the largest singular value of the $i^{th}$ conventional layer, estimated using a single power iteration update [57, 58]. Here, $\lambda$ controls to the level of smoothness imposed by $\mathcal{L}_{SR}$.

**More Expressive Approximate Posteriors with Normalizing Flows:** In NVAE, $p(z)$ and $q(z|x)$ are modeled by autoregressive distributions among groups and independent distributions in each group. This enables us to sample from each group in parallel efficiently. But, it also comes with the cost of less expressive distributions. A simple solution to this problem is to apply a few additional normalizing flows to the samples generated at each group in $q(z|x)$. Since they are applied only in the encoder network, i) we can rely on the inverse autoregressive flows (IAF) [4], as we do not require the explicit inversion of the flows, and ii) the sampling time is not increased because of the flows.

# 4 Experiments

In this section, we examine NVAE on several image datasets. We present the main quantitative results in Sec. 4.1, qualitative results in Sec. 4.2 and ablation experiments in Sec. 4.3.

Table 1: Comparison against the state-of-the-art likelihood-based generative models. The performance is measured in bits/dimension (bpd) for all the datasets but MNIST in which negative log-likelihood in nats is reported (lower is better in all cases). NVAE outperforms previous non-autoregressive models on most datasets and reduces the gap with autoregressive models.

| Method | MNIST 28×28 | CIFAR-10 32×32 | ImageNet 32×32 | CelebA 64×64 | CelebA HQ 256×256 | FFHQ 256×256 |
|---|---|---|---|---|---|---|
| NVAE w/o flow | **78.01** | 2.93 | - | 2.04 | - | 0.71 |
| NVAE w/ flow | 78.19 | **2.91** | 3.92 | **2.03** | **0.70** | **0.69** |
| **VAE Models with an Unconditional Decoder** | | | | | | |
| BIVA [36] | 78.41 | 3.08 | 3.96 | 2.48 | - | - |
| IAF-VAE [4] | 79.10 | 3.11 | - | - | - | - |
| DVAE++ [20] | 78.49 | 3.38 | - | - | - | - |
| Conv Draw [42] | - | 3.58 | 4.40 | - | - | - |
| **Flow Models without any Autoregressive Components in the Generative Model** | | | | | | |
| VFlow [59] | - | 2.98 | - | - | - | - |
| ANF [60] | - | 3.05 | 3.92 | - | 0.72 | - |
| Flow++ [61] | - | 3.08 | **3.86** | - | - | - |
| Residual flow [50] | - | 3.28 | 4.01 | - | 0.99 | - |
| GLOW [62] | - | 3.35 | 4.09 | - | 1.03 | - |
| Real NVP [63] | - | 3.49 | 4.28 | 3.02 | - | - |
| **VAE and Flow Models with Autoregressive Components in the Generative Model** | | | | | | |
| $\delta$-VAE [25] | - | 2.83 | 3.77 | - | - | - |
| PixelVAE++ [35] | 78.00 | 2.90 | - | - | - | - |
| VampPrior [64] | 78.45 | - | - | - | - | - |
| MAE [65] | 77.98 | 2.95 | - | - | - | - |
| Lossy VAE [66] | 78.53 | 2.95 | - | - | - | - |
| MaCow [67] | - | 3.16 | - | - | 0.67 | - |
| **Autoregressive Models** | | | | | | |
| SPN [68] | - | - | 3.85 | - | 0.61 | - |
| PixelSNAIL [34] | - | 2.85 | 3.80 | - | - | - |
| Image Transformer [69] | - | 2.90 | 3.77 | - | - | - |
| PixelCNN++ [70] | - | 2.92 | - | - | - | - |
| PixelRNN [41] | - | 3.00 | 3.86 | - | - | - |
| Gated PixelCNN [71] | - | 3.03 | 3.83 | - | - | - |

## 4.1 Main Quantitative Results

We examine NVAE on the dynamically binarized MNIST [72], CIFAR-10 [73], ImageNet $32\times32$ [74], CelebA $64 \times 64$ [75, 76], CelebA HQ 256×256 [28], and FFHQ 256×256 [77] datasets. All the datasets except FFHQ are commonly used for evaluating likelihood-based generative models. FFHQ is a challenging dataset, consisting of facial images. We reduce the resolution of the images in FFHQ to 256×256. To the best of our knowledge, NVAE is the first VAE model trained on FFHQ.

We build NVAE using the hierarchical structure shown in Fig. 2 and residual cells shown in Fig. 3. For large image datasets such as CelebA HQ and FFHQ, NVAE consists of 36 groups of latent variables starting from $8 \times 8$ dims, scaled up to $128 \times 128$ dims with two residual cells per latent variable groups. The implementation details are provided in Sec. A in Appendix.

The results are reported in Table 1. NVAE outperforms the state-of-the-art non-autoregressive flow and VAE models including IAF-VAE [4] and BIVA [36] on all the datasets, but ImageNet, in which NVAE comes second after Flow++[61]. On CIFAR-10, NVAE improves the state-of-the-art from 2.98 to 2.91 bpd. It also achieves very competitive performance compared to the autoregressive models. Moreover, we can see that NVAE's performance is only slightly improved by applying flows in the encoder, and the model without flows outperforms many existing generative models by itself. This indicates that the network architecture is an important component in VAEs and a carefully designed network with Normal distributions in encoder can compensate for some of the statistical challenges.

## 4.2 Qualitative Results

For visualizing generated samples on challenging datasets such as CelebA HQ, it is common to lower the temperature of the prior to samples from the potentially high probability region in the model [62].

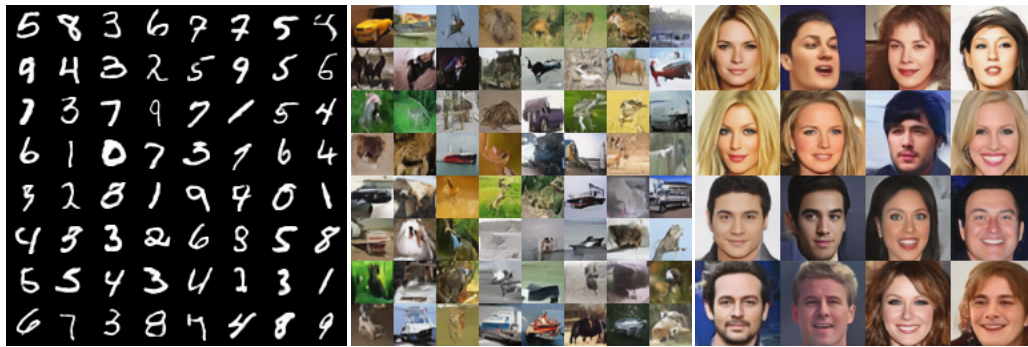

(a) MNIST ($t = 1.0$)       (b) CIFAR-10 ($t = 0.7$)       (c) CelebA 64 ($t = 0.6$)

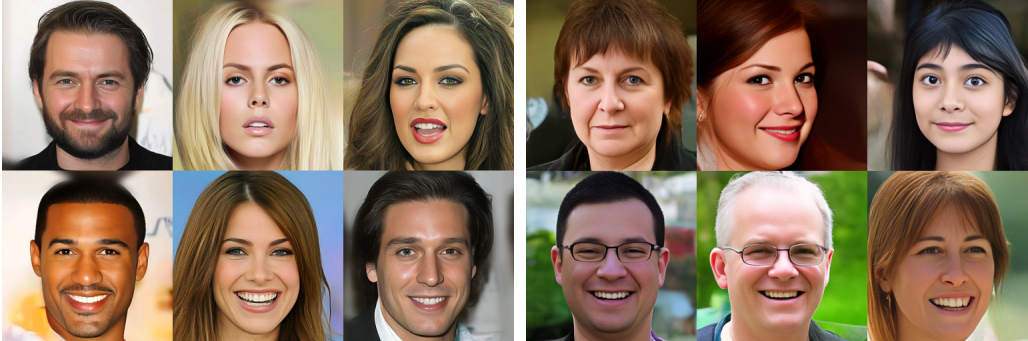

(d) CelebA HQ ($t = 0.6$)                 (e) FFHQ ($t = 0.5$)

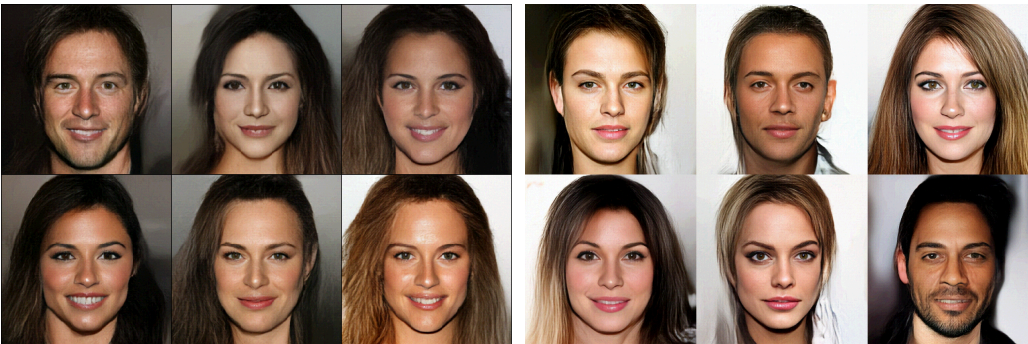

(f) MaCow [67] trained on CelebA HQ ($t = 0.7$)      (g) Glow [62] trained on CelebA HQ ($t = 0.7$)

Figure 4: (a)-(e) Sampled images from NVAE with the temperature in prior ($t$). (f)-(g) A few images generated by MaCow [67] and Glow [62] are shown for comparison (images are from the original publications). NVAE generates diverse high quality samples even with a small temperature, and it exhibits remarkably better hair details and diversity (best seen when zoomed in).

This is done by scaling down the standard deviation of the Normal distributions in each conditional in the prior, and it often improves the quality of the samples, but it also reduces their diversity.

In NVAE, we observe that if we use the single batch statistics during sampling for the BN layers, instead of the default running averages, we obtain much more diverse and higher quality samples even with small temperatures[3]. A similar observation was made in BigGAN [78] and DCGAN [79]. However, in this case, samples will depend on other data points in the batch. To avoid this, similar to BigGAN, we readjust running mean and standard deviation in the BN layers by sampling from the generative model 500 times for the given temperature, and then we use the readjusted statistics for the final sampling[4]. We visualize samples with the default BN behavior in Sec. B.2 in the appendix.

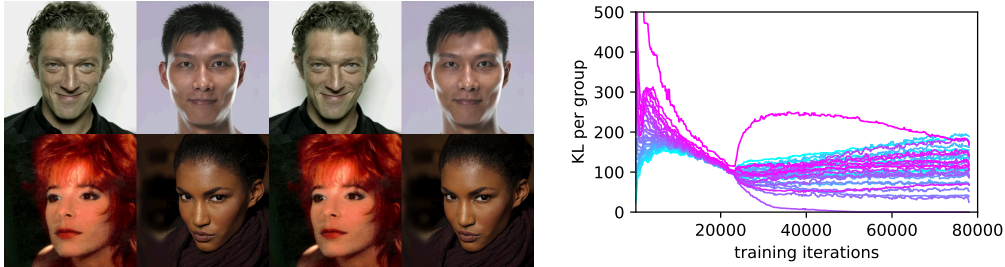

(a) Reconstruction results (best seen when zoomed in).      (b) Average KL per group.

Figure 5: (a) Input input on the left and reconstructed images on the right for CelebA HQ. (b) KL per group on CIFAR-10.

Fig. 4 visualizes the samples generated by NVAE along with the samples from MaCow [67] and Glow [62] on CelebA HQ for comparison. As we can see, NVAE produces high quality and diverse samples even with small temperatures.

## 4.3 Ablation Studies

In this section, we perform ablation experiments to provide a better insight into different components in NVAE. All the experiments in this section are performed on CIFAR-10 using a small NVAE, constructed by halving the number of channels in residual cells and removing the normalizing flows.

**Normalization and Activation Functions:** We examine the effect of normalization and activation functions on a VAE with cells visualized in Fig. 3b for different numbers of groups ($L$). ELU with WN and data-dependent initialization were used in IAF-VAE [4] and BIVA [36]. As we can see in Table 2, replacing WN with BN improves ELU's training, especially for $L = 40$, but BN achieves better results with Swish.

Table 2: Normalization & activation

| Functions | $L = 10$ | $L = 20$ | $L = 40$ |
|---|---|---|---|
| WN + ELU | 3.36 | 3.27 | 3.31 |
| BN + ELU | 3.36 | 3.26 | 3.22 |
| BN + Swish | **3.34** | **3.23** | **3.16** |

**Residual Cells:** In Table 3, we examine the cells in Fig 3 for the bottom-up encoder and top-down generative models. Here, "Separable" and "Regular" refer to the cells in Fig. 3a and Fig. 3b respectively. We observe that the residual cell with depthwise convolution in the generative model outperforms the regular cells, but it does not change the performance when it is in the bottom-up model. Given the lower memory and faster training with regular cells, we use these cells for the bottom-up model and depthwise cells for the top-down model.

Table 3: Residual cells in NVAE

| Bottom-up model | Top-down model | Test (bpd) | Train time (h) | Mem. (GB) |
|---|---|---|---|---|
| Regular | Regular | 3.11 | 43.3 | 6.3 |
| Separable | Regular | 3.12 | 49.0 | 10.6 |
| Regular | Separable | **3.07** | 48.0 | 10.7 |
| Separable | Separable | **3.07** | 50.4 | 14.9 |

**Residual Normal Distributions:** A natural question is whether the residual distributions improve the optimization of the KL term in the VAE objective or whether they only further contribute to the approximate posterior collapse. In Table 4, we train the 40-group model from Table 2 with and without the residual distributions, and we report the number of active channels in the latent variables[5], the average training KL, reconstruction loss, and variational bound in bpd. Here, the baseline without residual distribution corresponds to the parameterization used in IAF-VAE [4]. As we can see, the residual distribution does virtually not change the number of active latent variables or reconstruction loss. However, it does improve the KL term by 0.04 bpd in training, and the final test log-likelihood by 0.03 bpd (see Sec. B.4 in Appendix for additional details).

Table 4: The impact of residual dist.

| Model | # Act. $z$ | Training KL | Rec. | $\mathcal{L}_{\text{VAE}}$ | Test LL |
|---|---|---|---|---|---|
| w/ Res. Dist. | 53 | **1.32** | 1.80 | **3.12** | **3.16** |
| w/o Res. Dist. | 54 | 1.36 | 1.80 | 3.16 | 3.19 |

**The Effect of SR and SE:** In Table 5, we train the same 40-group model from Table 2 without spectral regularization (SR) or squeeze-and-excitation (SE). We can see that removing any of these components hurts performance. Although we introduce SR for stabilizing training, we find that it also slightly improves the generative performance (see Sec. B.5 in the appendix for an experiment, stabilized by SR).

Table 5: SR & SE

| Model | Test NLL |
|---|---|
| NVAE | **3.16** |
| NVAE w/o SR | 3.18 |
| NVAE w/o SE | 3.22 |

**Sampling Speed:** Due to the unconditional decoder, NVAE's sampling is fast. On a 12-GB Titan V GPU, we can sample a batch of 36 images of the size 256×256 px in 2.03 seconds (56 ms/image). MaCow [67] reports 434.2 ms/image in a similar batched-sampling experiment ($\sim 8\times$ slower).

**Reconstruction:** Fig. 5a visualizes the reconstruction results on CelebA HQ datasets. As we can see, the reconstructed images in NVAE are indistinguishable from the training images.

**Posterior Collapse:** Since we are using more latent variables than the data dimensionality, it is natural for the model to turn off many latent variables. However, our KL balancing mechanism (Sec. A) stops a hierarchical group from turning off entirely. In Fig. 5b, we visualize KL per group in CIFAR-10 (for 30 groups). Note how most groups obtain a similar KL on average, and only one group is turned off. We apply KL balancing mechanism only during KL warm-up (the first $\sim 25000$ iterations). In the remaining iterations, we are using the original VAE objective without any KL balancing (Eq. 1).

## 5    Conclusions

In this paper, we proposed Nouveau VAE, a deep hierarchical VAE with a carefully designed architecture. NVAE uses depthwise separable convolutions for the generative model and regular convolutions for the encoder model. We introduced residual parameterization of Normal distributions in the encoder and spectral regularization for stabilizing the training of very deep models. We also presented practical remedies for reducing the memory usage of deep VAEs, enabling us to speed up training by $\sim 2\times$. NVAE achieves state-of-the-art results on MNIST, CIFAR-10, CelebA 64, and CelebA HQ-256, and it provides a strong baseline on FFHQ-256. To the best of our knowledge, NVAE is the first VAE that can produce large high-quality images and it is trained without changing the objective function of VAEs. Our results show that we can achieve state-of-the-art generative performance by carefully designing neural network architectures for VAEs. The future work includes scaling up the training for larger images, experimenting with more complex normalizing flows, automating the architecture design by neural architecture search, and studying the role of batch normalization in VAEs. We have released our source-code to facilitate research in these directions.

## Impact Statement

This paper's contributions are mostly centered around the fundamental challenges in designing expressive neural architectures for image VAEs, and the ideas, here, are examined on commonly used public datasets. This work has applications in content generation, computer graphics, data augmentation, semi-supervised learning, and representation learning.

VAEs are known to represent the data distribution more faithfully than commonly used generative adversarial networks (GANs), as VAEs do not suffer from the mode collapse problem. Thus, in the long run, enabling VAEs to generate high-quality images will help us reduce bias in the generated content, produce diverse output, and represent minorities better.

One should also take into consideration that VAEs are trained to mimic the training data distribution, and, any bias introduced in data collection will make VAEs generate samples with a similar bias. Additional bias could be introduced during model design, training, or when VAEs are sampled using small temperatures. Bias correction in generative learning is an active area of research, and we recommend the interested readers to check this area [80] before building applications using this work.

## Acknowledgements

The authors would like to thank Karsten Kreis and Margaret Albrecht for providing feedback on the early version of this work. They also would like to extend their sincere gratitude to Sangkug Lym for providing suggestions for accelerating NVAE. Last but not least, they are grateful to Sabu Nadarajan, Nithya Natesan, Sivakumar Arayandi Thottakara, and Jung Seok Jin for providing compute support.

## Footnotes

[1]A $k \times k$ regular convolution, mapping a $C$-channel tensor to the same size, has $k^2 C^2$ parameters and computational complexity of $O(k^2 C^2)$ per spatial location, whereas a depthwise convolution operating in the same regime has $k^2 C$ parameters and $O(k^2 C)$ complexity per location.

[2]Swish cannot be done in place and it requires additional memory for the backward pass.

[3]For the evaluation in Sec. 4.1, we do use the default setting to ensure that our reported results are valid.

[4]This intriguing effect of BN on VAEs and GANs requires further study in future work. We could not obtain the same quantitative and qualitative results with instance norm which is a batch-independent extension to BN.

[5]To measure the number of the active channels, the average of KL across training batch and spatial dimensions is computed for each channel in latent variables. A channel is considered active if the average is above 0.1.

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
