[Supplementary Material]

# A  Additional Implementation Details

**Warming-up the KL Term:** Similar to the previous work, we warm-up the KL term at the beginning of training [43]. Formally, we optimize the following objective:

$$\mathbb{E}_{q(\boldsymbol{z}|\boldsymbol{x})}\left[\log p(\boldsymbol{x}|\boldsymbol{z})\right] - \beta \text{KL}(q(\boldsymbol{z}|\boldsymbol{x})||p(\boldsymbol{z})),$$

where $\beta$ is annealed from 0 to 1 at the first 30% of training.

**Balancing the KL Terms:** In hierarchical VAEs, the KL term is defined by:

$$\text{KL}(q(\boldsymbol{z}|\boldsymbol{x})||p(\boldsymbol{z})) = \sum_{l=1}^{L} \mathbb{E}_{q(\boldsymbol{z}_{<l}|\boldsymbol{x})}\left[\text{KL}(q(\boldsymbol{z}_l|\boldsymbol{x}, \boldsymbol{z}_{<l})||p(\boldsymbol{z}_l|\boldsymbol{z}_{<l}))\right],$$

where each $\text{KL}(q(\boldsymbol{z}_l|\boldsymbol{x}, \boldsymbol{z}_{<l})||p(\boldsymbol{z}_l|\boldsymbol{z}_{<l}))$ can be thought as the amount of information encoded in the $l^{th}$ group. In deep hierarchical VAEs, during training, some groups of latent variables can easily become deactivated by matching the approximate posterior with the prior (i.e., posterior collapse). One simple solution is to use KL balancing coefficients [20, 66] to ensure that an equal amount of information is encoded in each group using:

$$\text{KL}(q(\boldsymbol{z}|\boldsymbol{x})||p(\boldsymbol{z})) = \sum_{l=1}^{L} \gamma_l \, \mathbb{E}_{q(\boldsymbol{z}_{<l}|\boldsymbol{x})}\left[\text{KL}(q(\boldsymbol{z}_l|\boldsymbol{x}, \boldsymbol{z}_{<l})||p(\boldsymbol{z}_l|\boldsymbol{z}_{<l}))\right].$$

The balancing coefficient $\gamma_l$ is set to a small value when the KL term is small for that group to encourage the model to use the latent variables in that group, and it is set a large value when the KL term is large. The KL balancing coefficients are only applied during the KL warm-up period, and they are set to 1 afterwards to ensure that we optimize the variational bound. DVAE++ [20] sets $\gamma_l$ proportional to $\mathbb{E}_{\boldsymbol{x}\sim\mathcal{M}}\left[\mathbb{E}_{q(\boldsymbol{z}_{<l}|\boldsymbol{x})}\left[\text{KL}(q(\boldsymbol{z}_l|\boldsymbol{x}, \boldsymbol{z}_{<l})||p(\boldsymbol{z}_l|\boldsymbol{z}_{<l}))\right]\right]$ in each parameter update using the batch $\mathcal{M}$. However, since we have latent variable groups in different scales (i.e., spatial dimensions), we observe that setting $\gamma_l$ proportional to also the size of each group performs better, i.e., $\gamma_l \propto s_l \, \mathbb{E}_{\boldsymbol{x}\sim\mathcal{M}}\left[\mathbb{E}_{q(\boldsymbol{z}_{<l}|\boldsymbol{x})}\left[\text{KL}(q(\boldsymbol{z}_l|\boldsymbol{x}, \boldsymbol{z}_{<l})||p(\boldsymbol{z}_l|\boldsymbol{z}_{<l}))\right]\right]$

**Annealing $\lambda$:** The coefficient of the smoothness loss $\lambda$ is set to a fixed value in $\{10^{-2}, 10^{-1}\}$ for almost all the experiments. We used $10^{-1}$ only when training was unstable at $10^{-2}$. However, on Celeb-A HQ and FFHQ, we observe that training is initially unstable unless for $\lambda \in \{1, 10\}$ which applies a very strong smoothness. For these datasets, we anneal $\lambda$ with exponential decay from 10 to a small value shown in Table. 6 in the same number of iterations that the KL coefficient is annealed. Note that the smoothness loss is applied to both encoder and decoder. We hypothesize that a sharp decoder may require a sharp encoder, causing more instability in training.

**Weight Normalization (WN):** WN cannot be used with BN as BN removes any scaling of weights introduced by WN. However, previous works have seen improvements in using WN for VAEs. In NVAE, we apply WN to any convolutional layer that is not followed by BN, e.g., convolutional layers that produce the parameters of Normal distributions in encoder or decoder.

**Inverse Autoregressive Flows (IAFs):** We apply simple volume-preserving normalizing flows of the form $\boldsymbol{z}' = \boldsymbol{z} + \boldsymbol{b}(\boldsymbol{z})$ to the samples generated by the encoder at each level, where $\boldsymbol{b}(\boldsymbol{z})$ is produced by an autoregressive network. In each flow operation, the autoregressive network is created using a cell similar to Fig. 3 (a) with the masking mechanism introduced in PixelCNN [41]. In the autoregressive cell, BN is replaced with WN, and SE is omitted, as these operations break the autoregressive dependency. We initially examined non-volume-preserving affine transformations in the form of $\boldsymbol{z}' = \boldsymbol{a}(\boldsymbol{z}) \odot \boldsymbol{z} + \boldsymbol{b}(\boldsymbol{z})$, but we did not observe any improvements. Similar results are reported by Kingma et al. [4] (See Table 3).

**Optimization:** For all the experiments, we use the AdaMax [81] optimizer for training with the initial learning rate of 0.01 and with cosine learning rate decay. For FFHQ experiments, we reduce the learning rate to 0.008 to further stabilize the training.

**Image Decoder $p(\boldsymbol{x}|\boldsymbol{z})$ :** For all the datasets but MNIST, we use the mixture of discretized Logistic distribution [70]. In MNIST, we use a Bernoulli distribution. Note that in all the cases, our decoder is unconditional across the spatial locations in the image.

**Evaluation:** For estimating log-likelihood on the test datasets in evaluation, we use importance weighted sampling using the encoder [11]. We use 1000 importance weighted samples for evaluation.

Table 6: A summary of hyperparameters used in training NVAE with additional information. $D^2$ indicates a latent variable with the spatial dimensions of $D \times D$. As an example, the MNIST model consists of 15 groups of latent variables in total, covering two different scales. In the first scale, we have five groups of $4 \times 4 \times 20$-dimensional latent variables (in the form of height×width×channel). In the second scale, we have 10 groups of $8 \times 8 \times 20$-dimensional variables.

| Hyperparamter | MNIST $28{\times}28$ | CIFAR-10 $32{\times}32$ | ImageNet $32{\times}32$ | CelebA $64{\times}64$ | CelebA HQ $256{\times}256$ | FFHQ $256{\times}256$ |
|---|---|---|---|---|---|---|
| # epochs | 400 | 400 | 45 | 90 | 300 | 200 |
| batch size per GPU | 200 | 32 | 24 | 16 | 4 | 4 |
| # normalizing flows | 0 | 2 | 2 | 2 | 4 | 4 |
| # latent variable scales | 2 | 1 | 1 | 3 | 5 | 5 |
| # groups in each scale | 5, 10 | 30 | 28 | 5, 10, 20 | 4, 4, 4, 8, 16 | 4, 4, 4, 8, 16 |
| spatial dims of $z$ in each scale | $4^2, 8^2$ | $16^2$ | $16^2$ | $8^2, 16^2, 32^2$ | $8^2, 16^2, 32^2, 64^2, 128^2$ | $8^2, 16^2, 32^2, 64^2, 128^2$ |
| # channel in $z$ | 20 | 20 | 20 | 20 | 20 | 20 |
| # initial channels in enc. | 32 | 128 | 192 | 64 | 30 | 30 |
| # residual cells per group | 1 | 2 | 2 | 2 | 2 | 2 |
| $\lambda$ | 0.01 | 0.1 | 0.01 | 0.1 | 0.01 | 0.1 |
| GPU type | 16-GB V100 | 16-GB V100 | 32-GB V100 | 16-GB V100 | 32-GB V100 | 32-GB V100 |
| # GPUs | 2 | 8 | 24 | 8 | 24[*] | 24[*] |
| total train time (h) | 21 | 55 | 70 | 92 | 94 | 160 |

[*] A smaller model with 24 initial channels instead of 32, could be trained on only 8 GPUs in the same time (with the batch size of 6). The smaller models obtain only 0.01 bpd higher negative log-likelihood on these datasets.

**Channel Sizes:** We only set the initial number of channels in the bottom-up encoder. When we downsample the features spatially, we double the number of channels in the encoder. The number of channels is set in the reverse order for the top-down model.

**Expansion Ratio $E$:** The depthwise residual cell in Fig. 3a requires setting an expansion ratio $E$. We use $E = 6$ similar to MobileNetV2 [46]. In a few cells, we set $E = 3$ to reduce the memory. Please see our code for additional details.

**Datasets:** We examine NVAE on the dynamically binarized MNIST [72], CIFAR-10 [73], ImageNet $32 \times 32$ [74], CelebA $64 \times 64$ [75, 76], CelebA HQ [28], and FFHQ $256{\times}256$ [77]. For all the datasets but FFHQ, we follow Glow [62] for the train and test splits. In FFHQ, we use 63K images for training, and 7K for test. Images in FFHQ and CelebA HQ are downsampled to $256 \times 256$ pixels, and are quantized in 5 bits per pixel/channel to have a fair comparison with prior work [62].

**Hyperparameters:** Given a large number of datasets and the heavy compute requirements, we do not exhaustively optimize the hyperparameters. In our early experiments, we observed that the larger the model is, the better it performs. We often see improvements with wider networks, a larger number of hierarchical groups, and more residual cells per group. However, they also come with smaller training batch size and slower training. We set the number of hierarchical groups to around 30, and we used two residual cells per group. We set the remaining hyperparameters such that the model could be trained in no more than about a week. Table. 6 summarizes the hyperparameters used in our experiments.

# B   Additional Experiments and Visualizations

In this section, we provide additional insights into NVAE.

## B.1   Is NVAE Memorizing the Training Set?

In VAEs, since we can compute the log-likelihood on a held-out set, we can ensure that the model is not memorizing the training set. In fact, in our experiments, as we increase the model capacity (depth and width), we never observe any overfitting behavior especially on the datasets with large images. In most cases, we stop making the model large because of the compute and training time

Samples          Retrieved Images from Training Set

Figure 6: Top retrieved images from the training set are visualized for samples generated by NVAE in each row. The generated instances do not exist in the training set (best seen when zoomed in).

considerations. However, since the images generated by NVAE are realistic, this may raise a question on whether NVAE memorizes the training set.

In Fig. 6, we visualize a few samples generated by NVAE and the most similar images from the training data. For measuring the similarity, we downsample the images by $4\times$, and we measure $L_2$ distance using the central crop of the images. Since images are aligned, this way we can compare images using the most distinct facial features (eyes, nose, and mouth). As we can see, the sampled images are not present in the training set.

## B.2 Changing the Temperature of the Prior in NVAE

It is common to lower the temperature of the prior when sampling from VAEs on challenging datasets. In Fig. 7, we examine different temperatures in the prior with different settings for the batch norm layers.

## B.3 Additional Generated Samples

In Fig. 8 and Fig. 9, we visualize additional generated samples by NVAE, trained on CelebA HQ. In these figures, we use higher temperatures ($t \in \{0.6, 0.7, 0.8, 0.9\}$), but we manually select the samples.

## B.4 More on the Impact of Residual Normal Distributions

Fig. 10 visualizes the total number of active channels in all latent variables during training. Here, we compare the residual Normal distributions against the model that predicts the absolute parameters of the Normal distributions in the approximate posterior. This figure corresponds to the experiment that we reported in Table. 4. As we can see, in the initial stage of training, the model without residual distributions turns off more latent variables.

Batch Norm Statistics From Training    Batch Norm Statistics Readjusted

t = 0.1

t = 0.3

t = 0.5

t = 0.6

t = 0.7

t = 0.8

t = 0.9

t = 1.0

Figure 7: Randomly sampled images from NVAE with different temperatures in the prior for the CelebA HQ dataset (best seen when zoomed in). In the batch normalization layers during sampling, we examine two settings: i) the default mode that uses the running averages from training (on the left), and ii) readjusted mode in which the running averages are re-tuned by sampling from the model 500 times with the given temperature (on the right). Readjusted BN statistics improve the diversity and quality of the images, especially for small temperatures.

Figure 8: Additional 256×256-pixel samples generated by NVAE, trained on CelebA HQ [28]. In this figure, we use higher temperatures ($t \in \{0.6, 0.7, 0.8, 0.9\}$), but we manually select the samples.

Figure 9: Additional 256×256-pixel samples generated by NVAE, trained on CelebA HQ [28]. In this figure, we use higher temperatures ($t \in \{0.6, 0.7, 0.8, 0.9\}$), but we manually select the samples.

Figure 10: The total number of active channels in $z$ is reported for two models with and without residual distributions. The model with residual distribution keeps more latent variables active in the KL warm-up phase (up to 8K iterations), and it achieves a better KL value at the end of the training (see Table. 4)

## B.5 Stabilizing the Training with Spectral Regularization

In our experiments, we came across many cases whose training was unstable due to the KL term, and it was stabilized by spectral regularization. Initially, instead of spectral regularization, we examined common approaches such as gradient clipping or limiting the parameters of the Normal distributions to a small range. But, none could stabilize the training without negatively affecting the performance. Fig. 11 shows an experiment on the FFHQ dataset. The training is stabilized by increasing the spectral regularization coefficient ($\lambda$) from 0.1 to 1.0.

Figure 11: An example experiment on the FFHQ dataset. All the hyper-parameters are identical between the two runs. However, training is unstable due to the KL term in the objective. We stabilize the training by increasing the spectral regularization coefficient $\lambda$.

## B.6 Long-Range Correlations

NVAE's hierarchical structure is composed of many latent variable groups operating at different scales. For example, on CelebA HQ $256 \times 256$, the generative model consists of five scales. It starts from a spatially arranged latent variable group of the size $8 \times 8$ at the top, and it samples from the hierarchy group-by-group while gradually doubling the spatial dimensions up to $128 \times 128$.

A natural question to ask is what information is captured at different scales. In Fig. 12, we visualize how the generator's output changes as we fix the samples at different scales. As we can see, the

global long-range correlations are captured mostly at the top of the hierarchy, and the local variations are recorded at the lower groups.

Figure 12: Where does our hierarchical model capture long-range correlations? NVAE on CelebA HQ consists of latent variable groups that are operating at five scales (starting from $8 \times 8$ up to $128 \times 128$). In each row, we fix the samples at a number of top scales and we sample from the rest of the hierarchy. As we can see, the long-range global structure is mostly recorded at the top of the hierarchy in the $8 \times 8$ dimensional groups. The second scale does apply some global modifications such as changing eyes, hair color, skin tone, and the shape of the face. The bottom groups capture mostly low-level variations. However, the lowest scale can still make some subtle long-range modifications. For example, the hair color is slightly modified when we are only sampling from the lowest scale in the last row. This is potentially enabled because of the large receptive field in our depthwise separable residual cell.