[Reviews · NeurIPS 2020]

Review 1

Summary and Contributions: This paper improved the quality of VAE through carefully designed network architectures along with a bag of tricks. They achieved high-quality generation of 256x256 facial images without adversarial learning. The design and training details may inspire the community in various tasks of image synthesis. ========= After rebuttal =========== I keep my score considering the authors' response and other reviews.

Strengths: 1) The proposed method achieved generating high-resolution realistic images without adversarial training. As far as I know, this is the first time for VAE to synthesize 256x256 images by merely focusing on carefully designing neural architectures. 2) The authors provide a large number of practical solutions to design and train deep encoder and decoders. It may help us understanding how to improve the performance of neural networks through carefully designing.

Weaknesses: 1) Since this work aims to improve the performance of VAE, it is suggested to provide visual results of image reconstruction. It is noted that the neural architecture and posterior distribution have been changed dramatically compared to most VAEs. I wonder the reconstruction quality of the input images, and how it would change when sampling randomly from the posterior distribution. 2) Though this work focused on VAE, I think it is better to include GAN-based works in the background or related works when referring to generative models. Some recent SOTA GANs and hybrid methods with VAE, such as VAE/GAN[1], IntroVAE[2] and StyleALAE[3], are suggested to discussed briefly. [1] Autoencoding beyond pixels using a learned similarity metric. In ICML, 2016. [2] IntroVAE: Introspective Variational Autoencoders for Photographic Image Synthesis. In NeurIPS, 2018. [3] Adversarial Latent Autoencoders. In CVPR, 2020. 3) One of the major differences of NVAE compared to other VAEs is the design of the latent code. The latent dimension is increased significantly and modeled as dependent distributions via autoregressive techniques. It is suggested to analyze deeply the influence of such design. As well known, the original VAE is able to learn somewhat disentangling representations, which is helpful to improve the interpretability and controllability of image synthesis. I think it may further improve the quality of this work by discussing the effect of the non-dimension-reduction and non-disentangling design theoretically or empirically. 4) I wonder whether NVAE works well on non-face images of high-resolution. Most SOTA GANs are able to produce high-resolution natural images on ImageNet database (conditioned on class labels or successful on some specific classes). Therefore, it is worth investigating the possibility of generating high-fidelity results of natural images.

Correctness: Yes.

Clarity: Yes.

Relation to Prior Work: Yes.

Reproducibility: Yes

Additional Feedback:


Review 2

Summary and Contributions: The paper proposes a bidirectional hierarchical VAE architecture, that couples the prior and the posterior via a residual parametrization and a combination of training tricks, and achieves sota results among non-autoregressive, latent variable models on natural images. The final, however, predictive likelihood achieved is almost identical (2.91 vs 2.92) compared to that of a pixel-autoregressive purely generative model. According to my understanding, the "autoregression" is achieved in the latent space as opposed to the pixel space. The dimension of the latent variables needed for the reconstruction is quite large (larger than the input space), hence compromising the compression capability of the model.

Strengths: The paper provides exhaustive experimental analysis and ablation studies for every architectural and training choices. The choices are empirically validated. The most important, according to my opinion, contribution is the introduction of the residual distributions for effectively parametrizing the posterior through some residual, data-dependent, mean and variance terms of the prior. Finally, the quality of the generated images is quite impressive.

Weaknesses: Some points in the proposed architecture are still vague to me. For example, 1) is the data layer conditioned on the z-s of all the layers or only on the last one? according to equation (1)? 2) how are the \sigma-s parametrized? do the authors follow the same parametrization as in the IAF paper? is the \sigma bounded in [0,1]? 3) could the authors please provide the learning curves to demonstrate training stability? moreover, it would be very helpful if the KL per layer was also reported to demonstrate that there is no posterior collapse. 4) in the results reported in Table 1, it is not clear whether the authors train an importance weighted vae or not, for fair comparison with the rest of the models. moreover?

Correctness: the empirical methodology is correct, but I feel that some details, as previously described, and the learning curves are missing.

Clarity: The paper was very well-written and easy to follow.

Relation to Prior Work: yes. the main difference is the use of the residual distributions which cooperate well with other existing techniques, such as batch normalization, the swish activation and spectral regularization

Reproducibility: Yes

Additional Feedback: 1) Do the authors believe that using a pixelcnn++ decoder could further push down the bits per dimension? how well, would the proposed latent variable hierarchy would work with an autoregressive decoder? 2) could the authors provide an explanation why the mnist model with flows performs worse, as seen in table 1? 3) given that the marginal likelihood achieved is almost the same with that of a purely generative pixel autoregressive model (pixel-cnn++), I think a comparison with the computational time needed for pixel-cnn++ would strengthen the contribution. Otherwise, it seems that the model just shifts the load from the pixel-space to the latent-space to achieve almost the same reconstruction. Post-rebuttal comment: I appreciate the authors' response. Below are just some suggestions, for a revised version of the paper, that I think would help other researchers in the area: i) for training stability, I think it would help if the authors also provide not just the KL as in Fig 10 in the supplementary, but also the full loss- regularized-negative ELBO. ii) Could the authors also report the amount of the posterior collapse, in order for the reader to get a sense of the redundancy in the latent space / compression achieved by the model? iii) in terms of the computational benefits of NVAE over PixelCNN++, it would be helpful if the *training*, not only the sampling, time was also reported.


Review 3

Summary and Contributions: The paper presents Nouveau VAE, a deep hierarchical VAE with a novel architecture consisting of 1. depthwise separabale convs to increase receptive field of generator without introducing lots of params, and batch norm, swish activation and squeeze excitation in architecture of residual block to further improve performance 2. stabilise optimisation of the KL by using spectral regularisation and a residual parametrisation for the posterior gaussians. They obtain SOTA results for non-autoregressive likelihood based models on image generation tasks, and are the first VAE to successfully produce good samples for images at 256x256 resolution. == Post rebuttal update == The authors addressed the reproducibility by promising to release code with instructions. I didnt deduct score for this so I maintain my score of 8.

Strengths: - SOTA quantitative benchmarks among non-autoregressive models, bridging gap with AR models while being fast to sample due to no autoregressive components in decoder. The closest model is ANF in Table 1, maybe you could include a discussion about it in related work. - First high resolution (256x256) samples of high quality from a VAE (except VQ-VAE, but that requires two stage training, doesn’t obtain comparable log-likelihood and is slow to sample from due to autoregressive prior). Samples on CelebA-HQ also have high diversity due to BN trick The result is significant to the community in showing that VAE’s can be made competitive if we tune the architecture properly. The empirical evaluation is sound with clear ablations showing each of the novel components brings improvements.

Weaknesses: - Seems to need lots of tuning to optimise well - We need spectral regularisation, residual parametrisation and in supplementary, there’s further tricks like warming up KL, balancing KL terms and annealing \lambda for spectral regularisation. This could make it hard to reproduce, and simplifying some choices might help wider adoption. (though if code is released this would be alleviated) - BN statistics readjusting seems a sampling hack - It would be nice to have a table that shows what is its effect on test set likelihood, and also to have a more theoretical justification for why its ok to do.

Correctness: Yes, claims are well justified by the results, and empirical methodology is correct.

Clarity: Paper is easy to understand and notation is clear.

Relation to Prior Work: Yes, discussion of prior work clearly distinguishes this work from similar work like BIVA, IAF-VAE’s, VQ-VAE. Also like the intro summarising a lot of the progress and learnings from other related vae work.

Reproducibility: Yes

Additional Feedback:


Review 4

Summary and Contributions: This paper proposes a reimplementation of the standard VAE algorithm with improved architectural designs. The proposed implementation achieves SOTA results on cifar10 wrt bpd among VAEs that do not have spatial autoregressive priors. Samples produced by the model is also very impressive, and qualitatively better than VAEs of similar kind. ---------------------------update after rebuttal---------------------- I have read the author's response and decide to maintain my rating.

Strengths: re1. the paper provides a very good implementation of VAE which achieves competitive results. 2. experiments are very extensive with explanations of detailed design choices 3. the paper is well written

Weaknesses: The main weakness is lack of novelty, in the sense that it is hard to distill a single new idea that addresses the limitation of previous works. But I do understand that the nature of the work is more focused on implementation details, which is a valuable contribution on its own as well.

Correctness: The claims and results are generally sound. However, I'm not convinced that the proposed depth wise separable conv is actually beneficial. Reading from Table 4, it seems that separable achieves better results at the cost of more compute and memory consumption. It'd be great if you can show the benefit of separable conv by equating the two factors.

Clarity: The paper is in general well written and reads smoothly.

Relation to Prior Work: The description about VQVAE is not entirely accurate. In fact, the original VQ-VAE paper https://arxiv.org/abs/1711.00937 shows that you can derive of lower bound of log likelihood and it is actually still following the VAE objective.

Reproducibility: Yes

Additional Feedback:

[Author Response · NeurIPS 2020]



(a) Reconstruction results (best seen when zoomed in).

(b) Average KL per group.

Figure 1: (a) Input on the left and reconstructed image on the right for CelebA HQ 256. (b) KL per group on CIFAR10.

We would like to thank all the reviewers for positive and constructive feedback.

**R1** ————————————————————————————————————————

**Reconstruction:** The reconstructed images in NVAE are indistinguishable from the training images (see Fig. 1(a)).

**Discussing additional SOTA hybrid models:** Thanks for pointing this out. We will include this in our final version

**Non-dimension-reduction and non-disentangling design:** Since our goal is to maximize (a lower bound on) the
marginal data log-likelihood, hierarchical dependencies help us build expressive approximate posteriors, which often
result in better generative performance. However, when the training goal shifts towards disentangled representation
learning, we can sacrifice generative performance for representation learning (See $\beta$-VAE, "Fixing a Broken ELBO").

**Non-face datasets such as ImageNet:** We haven't explored the ImageNet dataset beyond the 32x32 version. However,
our hypothesis is that ImageNet may correspond to a high-entropy distribution (compared to face images) which may
require even bigger VAE models. GANs are perhaps less prone to this, as they may drop modes without being penalized.

**R2** ————————————————————————————————————————

**Is the data conditioned on all $z$'s:** Yes, because the representation at the bottom of the top-down model (Fig. 2(b)) is
a function of all $z$'s.

**Parametrization for $\sigma$:** We parameterize $\sigma$'s in their log space, and we limit $\log \sigma$ to be in [-5, 5] which is much larger.

**Training curves:** Fig. 10 in the supplementary material demonstrates training stability with spectral regularization.

**Posterior collapse:** Since we are using more latent variables than the data dimensionality, it is natural for VAE objective
to turn off many latent variables. However, our KL balancing mechanism (Sec. A in the appendix) stops the hierarchical
groups from turning off. In Fig. 1(b), you can see KL per group in CIFAR10 (for 30 groups). Note how most groups
obtain a similar KL on average, and only one group is turned off. We apply KL balancing mechanism only during KL
warm-up (the first $\sim$ 25000 iterations). In the remaining, we are using ELBO without any KL balancing (Eq. 1).

**Importance weighted (IW) AE:** We do not train our model with the IW bound. The comparison to the previous work
is fair in this regard as we also use ELBO for training (see Eq. 1).

**Autoregressive decoder + NVAE:** Autoregressive decoders 1) are often slow to sample from and 2) tend to turn off
latent variables. A careful study is required in this space.

**Sampling time compared to pixelCNN++:** On CIFAR10 using a Titan V GPU and batch size of 16, NVAE takes 5.1
ms per image vs. 8,898.75 ms per image for pixelCNN++. NVAE is $\sim$1700x faster than pixelCNN++.

**R3** ————————————————————————————————————————

**Reproducibility:** Our current top priority is to release the code publicly with clear instructions to reproduce the results.

**Test likelihood after BN adjustment:** Thanks for suggesting this. Since the generative model changes after re-
adjusting the BN layers, this requires encoder retraining. We'll add a careful examination of this to the final version.

**R4** ————————————————————————————————————————

**Lack of novelty:** NVAE offers expressive networks and it enables stable training of deep hierarchical VAEs for large
images. We believe NVAE has the potential of becoming the backbone of future VAEs with complex statistical models.

**Efficacy of depthwise conv:** Depthwise conv has less computational complexity, however, it is still slow on current
hardware because of memory bottleneck. We agree with you that depthwise conv seems less efficient, but in our early
experiments, the generative quality of regular conv could not match it, even with more channels or longer training.

**VQ-VAE:** Thanks for pointing this out. We will update our description of VQ-VAE to indicate that it was originally
motivated by deriving a lower bound on log-likelihood.

[Meta-Review · NeurIPS 2020]

All four viewers provide favorable or very favorable reviews. The reviewers point out that the clear presentation and impressive empirical results. The paper is therefore accepted for a spotlight.